# A One-layer Neural Network for Robust Mean-Variance Portfolio Selection Problem

Keying Zhou
*School of Mathematics and Statistics*
*Chongqing Jiaotong University*
Chongqing, China
622220150002@mails.cqjtu.edu.cn

Jin Hu*
*School of Mathematics and Statistics*
*Chongqing Jiaotong University*
Chongqing, China
jhu@cqjtu.edu.cn

*Abstract*—Driven by globalization and technological innovation, the financial markets have experienced unprecedented volatility and uncertainty. Portfolio selection is a fundamental strategy in finance. During recent decades, the frequent occurrence of extreme market events and uncertainties has exposed significant limitations in the traditional mean-variance model, emphasizing the critical need for more robust approaches. In response to these challenges, this paper introduces a neurodynamic approach to robust portfolio selection. This approach is capable of efficiently handling high-dimensional data through massively parallel processing, providing a resilient solution to the complexities of modern financial markets. First, the corresponding robust counterpart model is derived by eliminating uncertainty from the robust portfolio selection model under the box uncertainty set. Consequently, the robust portfolio selection problem is transformed into a solvable quadratic programming problem. Next, a one-layer neural network model is constructed based on the Karush-Kuhn-Tucker (KKT) conditions. Subsequently, the stability and convergence of the proposed neural network are analyzed. Finally, simulation experiments are conducted using two global stock market datasets. The proposed neural network model demonstrates convergence even with large-scale data in the second dataset, highlighting the effectiveness of the neurodynamic approach in addressing robust portfolio selection problems.

*Index Terms*—Robust Portfolio Selection, Box Uncertainty Set, Neurodynamic Approach, Markowitz Mean-Variance Model.

## I. INTRODUCTION

Portfolio selection involves allocating capital across various assets to maximize returns while minimizing risk. The traditional mean-variance optimization framework, introduced by Markowitz [1], has long been a fundamental approach to efficient portfolio construction. However, market volatility and unpredictability frequently impair investors' ability to accurately forecast asset returns and risks. The emergence of robust optimization approaches has revolutionized portfolio management by addressing the limitations of conventional methods and incorporating uncertainty into the decision-making process. Robust optimization has garnered significant attention in recent years as a modeling framework for managing uncertainty in mathematical optimization. The foundation

The work described in this paper was supported by the National Natural Science Foundation of China under Grant 62176032 and 62276034, Joint Training Base Construction Project for Graduate Students in Chongqing under Grant JDLHPYJD2021016, Group Building Scientific Innovation Project for universities in Chongqing under Grant CXQT21021.
    *Corresponding author.

for robust optimization was established by the seminal work of Soyster [2], and further advanced by the pivotal contributions of Ben-Tal and Nemirovski [3], [4], and El Ghaoui and Lebret [5]. These works not only formalized the mathematical framework of robust optimization but also inspired numerous subsequent studies. In particular, robust portfolio selection has evolved alongside the incorporation of uncertainty [6]–[8]. Swain and Ojha [9] discussed robust mean-variance and robust mean-semivariance problems under box uncertainty. Hosseini-Nodeh et al. [10] examined a distributionally robust portfolio selection problem with fuzzy stochastic dominance constraints, assuming an unknown distribution of asset returns. Compared to traditional shrinkage-based and constrained portfolios, Petukhina [11] demonstrated that robustified portfolios have the lowest turnover while maintaining or slightly improving out-of-sample performance. Ai et al. [12] proposed a differential evolutionary algorithm to optimize asset allocation and maximize returns subject to risk constraints in portfolio optimization. As financial markets continue to evolve, it is evident that robust portfolio selection plays a crucial role in both theory and practice, offering protection and growth potential for investors.

Uncertainty sets are a critical component of robust optimization, as their selection directly influences the complexity of the problem. For instance, the robust counterpart of an uncertain linear programming problem under box or polyhedral uncertainty sets remains a linear programming problem [13]. Conversely, the robust counterpart of an uncertain linear programming problem under ellipsoidal uncertainty sets is a second-order conic programming model (SOCP) [14]. Under single-ellipsoidal uncertainty sets, the robust counterpart of uncertain quadratically constrained quadratic programming (QCQP) and second-order cone programming (SOCP) problems are semidefinite programming (SDP) problems [15]. Moreover, the robust counterpart of the QCQP and SOCP problems under the intersection of polyhedral or ellipsoidal uncertainty sets is NP-hard [16], indicating significant computational complexity. Given these considerations, this paper adopts a more practical approach by selecting box uncertainty sets to simplify the problem. This approach not only reduces computational complexity, but also simplifies the problem, making it more straightforward to handle and providing a

feasible solution for practical applications.

Theoretically appealing as it is, robust portfolio selection poses significant computational challenges. Choosing an appropriate set of uncertainties to address the optimization problem requires substantial computational power and a sophisticated process. As the number of asset classes in the portfolio increases, the dimension of the optimization problem also increases, leading to increased computational costs and potential scalability issues. Furthermore, accurately estimating the uncertainty set becomes even more challenging when historical data is insufficient or market dynamics are rapidly evolving. Consequently, integrating advanced computational techniques into the research process is crucial to addressing these challenges.

Neurodynamic optimization offers a promising approach by combining the principles of artificial neural networks and dynamical systems theory to resolve complex optimization problems. The parallel processing capability of neural networks reduces the computational burden of traditional robust optimization methods and scales efficiently with robust portfolio selection sizes, making them an optimal choice in volatile markets. The pioneering work of Hopfield and Tank [17], [18] established the foundational framework for using neural networks as content-addressable memory systems, which was later extended to optimization fields such as nonlinear programming [19], nonsmooth programming [20], nonconvex programming [21], biconvex programming [22], multi-objective programming [23], and interval-valued optimization [24]. Xia et al. [25] provided a comprehensive overview of neurodynamic optimization, summarizing recent advances in model structure, convergence properties, and solvability range.

The neurodynamic approach excels in portfolio selection. Liu et al. [26] presented a one-layer recurrent neural network for solving pseudo-convex optimization problems with linear equality and constraints and discussed applications to dynamic portfolio selection optimization. Subsequently, Leung and Wang [27] proposed a collaborative neurodynamic optimization approach for cardinality-constrained portfolio selection. Leung et al. [28] explored portfolio selection based on neurodynamic optimization, demonstrating its superiority over three benchmark approaches in terms of risk-adjusted performance criteria and portfolio returns. Cao et al. [29] developed a recurrent neural network model to provide a rigorous theoretical analysis of convergence and optimality in portfolio optimization. Cao and Li [30] proposed three novel dynamic neural networks to address nonconvex portfolio optimization in the presence of transaction costs and quantitative constraints. Additionally, the neurodynamic approach has achieved significant results in robotics, engineering, and technology [31]–[33].

In 2023, Hu et al. [34] studied robust linear programming under polyhedral uncertainty sets using a neurodynamic approach, achieving notable results. The research was not only innovative in theory but also demonstrated considerable potential for practical applications. Their work furnished us with a robust theoretical foundation and methodological guidance.

Building upon this foundation, we apply the neurodynamic approach to address the robust portfolio selection problem. Furthermore, we have selected box uncertainty sets for analysis. Box uncertainty sets define uncertainty by specifying upper and lower bounds for variables, which are easier to adjust than polyhedral uncertainty sets. This flexibility allows quick model updates based on new information or requirements, reducing computational complexity.

The contributions of this paper are summarized as follows.

1) Based on the Markowitz Mean-Variance portfolio selection optimization model, we consider the box uncertainty set and study the uncertainty in the expected return of an asset, which provides a new perspective in dealing with uncertainty.
2) The stability and convergence of the proposed neural network are analyzed in theory.
3) Compared with existing optimization approaches, the proposed method does not require high computational power and is therefore suitable for dealing with real-world problems with uncertainty.

The remainder of this paper is organized as follows. Section II introduces preliminaries. Section III presents the main results. Section IV presents stability analysis. Section V discusses simulation results. Section VI concludes this paper.

## II. PRELIMINARIES

**Definition 1.** [35] *(Set-valued map) Suppose that to each point $x$ of a set $\mathcal{E} \subset \mathbb{R}^n$, there corresponds a nonempty set $F(x) \subset \mathbb{R}^n$, then $x \to F(x)$ is a set-valued map from $\mathcal{E}$ to $\mathbb{R}^n$. A set-valued map $F: \mathcal{E} \to \mathbb{R}^n$ with nonempty values is said to be upper semicontinuous at $x_0 \in \mathcal{E}$ if for any open set $\mathcal{V}$ containing $F(x_0)$, there exists a neighborhood $\mathcal{U}$ of $x_0$ such that $F(\mathcal{U}) \subset \mathcal{V}$. If $\mathcal{E}$ is closed, $F$ has nonempty closed values, and it is bounded in a neighborhood of each point $x \in \mathcal{E}$, then $F$ is upper semicontinuous on $\mathcal{E}$ if and only if its graph $\{(x, y) \in \mathcal{E} \times R^n : y \in F(x)\}$ is closed.*

**Definition 2.** [36] *In the following dynamical system:*

$$\dot{u} = \Phi(u), \quad u(0) = u_0 \in \mathbb{R}^n, \tag{1}$$

*where $\Phi$ is a function from $\mathbb{R}^n \to \mathbb{R}^n$, $\hat{u}$ is said to be an equilibrium point of the above dynamical system if $\Phi(\hat{u}) = 0$.*

**Definition 3.** [37] *Let $u(t)$ be a solution trajectory of a system $\dot{u} = F(t, u)$, and let $\Omega^*$ denotes the set of equilibrium points of this equation. The solution trajectory is said to be globally convergent to the set $\Omega^*$, if $u^*$ satisfies*

$$\lim_{t \to \infty} \text{dist}(u(t), \Omega^*) = 0, \tag{2}$$

*where $\text{dist}(u(t), \Omega^*) = \inf_{v \in \Omega^*} \|u - v\|$. In particular, if the set $\Omega^*$ has only one point $u^*$, then $\lim_{t \to \infty} u(t) = u^*$, and the system $\dot{u} = F(t, u)$ is said to be globally asymptotically stable at $u^*$ if the system is also stable at $u^*$ in the sense of Lyapunov.*

**Definition 4.** [38] *A variational inequality* $\mathrm{VI}(F,\Omega)$ *for* $F : \Omega \subset \mathbb{R}^l \to \mathbb{R}^l$ *is to find* $u^* \in \Omega$ *such that*

$$(u - u^*)^{\mathrm{T}} F(u^*) \geq 0, \quad \forall u \in \Omega. \tag{3}$$

*In the special case where $F$ is affine and $\Omega$ is the nonnegative orthant, this problem reduces to the classical linear complementarity problem.*

**Definition 5.** [39] *A function $\psi : \mathbb{R}^m \to \Omega \subset \mathbb{R}^m$ is said to be Lipschitz continuous with constant $L$ on a set $\Omega$ if, for each pair of points $u, v \in \Omega$*

$$\|\psi(u) - \psi(v)\| \leq L\|u - v\| \tag{4}$$

*where $\| \cdot \|$ denotes the $l_2$ norm of $\mathbb{R}^m$. $\psi$ is said to be locally Lipschitz continuous on $\Omega$ if each point of $\Omega$ has a neighborhood $D_0 \subset \Omega$ such that the above inequality holds for each pair of points $u, v \in D_0$.*

**Lemma 1.** [40] *(Chain Rule) If $V(x) : \mathbb{R}^n \to \mathbb{R}$ is regular and $x(t) : [0, +\infty) \to \mathbb{R}^n$ is absolutely continuous on any compact interval of $[0, +\infty)$, then $x(t)$ and $V(x(t)) : [0, +\infty) \to \mathbb{R}$ are differentiable, and*

$$\dot{V}(x(t)) = \langle \xi, \dot{x}(t) \rangle \quad \forall \xi \in \partial V(x(t)) \tag{5}$$

*for a.e. $t \in [0, +\infty)$ .*

**Lemma 2.** [41] *If the Hessian matrix of $f$, $\nabla^2 f(x)$ is positive definite, then the gradient of $f$ is strictly monotone.*

**Lemma 3.** [41] *If a mapping $G$ is continuously differentiable on an open convex set $\mathcal{C}$ including $\Omega$ , then $G$ is monotone (strictly monotone and strongly monotone) on $\Omega$ if and only if its Jacobian matrix $\nabla G(x)$ is positive semidefinite (positive definite and uniformly positive definite) for all $x \in \Omega$ .*

**Lemma 4.** [42] *For any $u \in \mathbb{R}^n$ and $v \geq 0$, the following inequality holds:*

$$(u - v^+)^{\mathrm{T}}(u^+ - v) \geq 0, \tag{6}$$

*where $(u)^+ = [(u_1)^+, \ldots, (u_n)^+]^{\mathrm{T}}$, $(u_i)^+ = \max\{u_i, 0\}$ , and $i = 1, \ldots, n$.*

**Lemma 5.** [43] *(LaSalle invariant set theorem) Consider the system of the form $\dot{u} = F(t, u)$, with $F$ continuous and let $V(u)$ be a scalar function with continuous first partial derivatives. Assume that:*

(1) *for some $l > 0$, the region $\Omega_l$ defined by $V(u) < l$ is bounded,*

(2) *$\dot{V}(u) \leq 0$ for all $u \in \Omega_l$.*

*Let $\mathfrak{R}$ be the set of all points within $\Omega_l$ where $\dot{V}(u) = 0$, and $\mathcal{L}$ be the largest invariant set in $\mathcal{L}$. Then every solution $u(t)$ originating in $\Omega_l$ tends to $\mathcal{L}$ as $t \to \infty$.*

## III. MAIN RESULTS

In this paper, we assume that uncertainty exists only in the expected returns and there is no uncertainty in the covariance. Let $\mu_i$ be the expected value of the return of the $i$ th asset and $\sigma_{ij}$ be the covariance of the return between the $i$ th and $j$ th

asset. Consider an asset $i$ with a return over a period of time $t$ as $r_{it}(t = 1, 2, .., T)$.

Under the premise that short is not allowed, the robust mean-variance portfolio problem is defined as:

$$
\begin{aligned}
\min \ & \frac{1}{2} x^{\mathrm{T}} \hat{\Sigma} x \\
\text{s.t. } & \tilde{\mu} x \geq \tau \\
& \mathbf{e}^{\mathrm{T}} x = 1 \\
& x \geq 0 \\
& \tilde{\mu} \in \mathcal{U},
\end{aligned} \tag{7}
$$

where $x \in \mathbb{R}^n$, $\hat{\Sigma}$ is the variance matrix, $\tau$ is a given level of expected portfolio return, $\mathbf{e}$ denotes a vector where each element is one, $\tilde{\mu} \in \mathbb{R}^n$ is the coefficient of uncertainty belonging to the uncertainty set $\mathcal{U}$ and

$$
\hat{\Sigma} = \begin{bmatrix}
\sigma_{11} & \sigma_{12} & \ldots & \sigma_{1n} \\
\sigma_{21} & \sigma_{22} & \ldots & \sigma_{2n} \\
\vdots & \vdots & \ddots & \vdots \\
\sigma_{n1} & \sigma_{n2} & \ldots & \sigma_{nn}
\end{bmatrix}.
$$

Suppose that $\mu^0$ is a nominal value vector of uncertain parameters. Then the box uncertainty set is

$$\mathcal{U} = \{\tilde{\mu} : |\tilde{\mu}_i - \mu_i^0| \leq \delta_i, i = 1, 2, \ldots, n.\}, \tag{8}$$

where $\delta_i$ is the perturbation of the uncertain parameters. In all the discussion that follows, this paper analyzes the problem based on the box uncertainty set $\mathcal{U}$.

In problem (7), the uncertain parameters exist in the following inequality constraints:

$$\tilde{\mu} x \geq \tau. \tag{9}$$

We can write the above equation (9) in equivalent component form:

$$\sum_{i=1}^{n} \tilde{\mu}_i x_i \geq \tau, \quad i = 1, 2, \ldots, n. \tag{10}$$

Applying the idea of robust optimization to the above equation (10) and eliminating the uncertain parameters from the equation, first we have

$$\min_{\tilde{\mu}_i \in \mathcal{U}} \sum_{i=1}^{n} \tilde{\mu}_i x_i \geq \tau, \quad i = 1, 2, \ldots, n. \tag{11}$$

At this point, problem (7) turns out to be:

$$
\begin{aligned}
\min \ & \frac{1}{2} \sum_{i=1}^{n} \sum_{j=1}^{n} \sigma_{ij} x_i x_j \\
\text{s.t. } & \min_{\tilde{\mu}_i \in \mathcal{U}} \sum_{i=1}^{n} \tilde{\mu}_i x_i \geq \tau \\
& \sum_{i=1}^{n} x_i = 1, \quad i = 1, 2, \ldots, n \\
& x_i \geq 0 \\
& \tilde{\mu}_i \in \mathcal{U}.
\end{aligned} \tag{12}
$$

As stated in [44], the aforementioned problem can be transformed into a bi-level optimization problem. The lower-level problem involving inequalities with uncertain terms can be reformulated using Karush-Kuhn-Tucker (KKT) conditions. This results in a bilevel optimization problem that can be transformed into a single-level optimization problem to be solved. The specific proof can be found in Theorem 3.1 of [44]. Therefore, according to [44], the robust counterpart of Problem (7) can be obtained as

$$
\begin{aligned}
\min \ & \frac{1}{2}x^{\mathrm{T}}\hat{\Sigma}x \\
\text{s.t. } & \tau - (\mu^0 - \delta)x \le 0 \\
& \mathbf{e}^{\mathrm{T}}x - 1 = 0 \\
& x \ge 0.
\end{aligned}
\tag{13}
$$

The Lagrangian function of problem (13) is defined as:

$$
L(x,y,z) = \frac{1}{2}x^{\mathrm{T}}\hat{\Sigma}x + y^{\mathrm{T}}[\tau - (\mu^0 - \delta)x] - z^{\mathrm{T}}(\mathbf{e}^{\mathrm{T}}x - 1), \tag{14}
$$

where $y$ and $z$ are Lagrange multipliers.

Based on the above derivation, we can obtain the KKT condition for problem (13):

$$
\begin{cases}
\nabla_x L = \hat{\Sigma}x + [-\mu^0 + \delta]^{\mathrm{T}}y - \mathbf{e}z = 0 \\
\tau - (\mu^0 - \delta)x \le 0 \\
\mathbf{e}^{\mathrm{T}}x - 1 = 0 \\
y^{\mathrm{T}}\left[\tau - (\mu^0 - \delta)x\right] = 0 \\
y \ge 0 \\
x \ge 0
\end{cases}
\tag{15}
$$

Let $w = (x,y,z)^{\mathrm{T}}$ and

$$
F(w) = \begin{pmatrix}
\hat{\Sigma}x + [-\mu^0 + \delta]^{\mathrm{T}}y - \mathbf{e}z \\
-(\tau - (\mu^0 - \delta)x) \\
\mathbf{e}^{\mathrm{T}}x - 1
\end{pmatrix}, \tag{16}
$$

then $w^* = (x^*, y^*, z^*)^{\mathrm{T}}$ is a KKT point if $w^*$ is also a solution of the following variational inequality problem:

$$
(w - w^*)^{\mathrm{T}}F(w^*) \ge 0, \forall w \ge 0. \tag{17}
$$

This can be equivalently written as

$$
\begin{cases}
(x - x^*)^{\mathrm{T}}[\hat{\Sigma}x^* + (-\mu^0 + \delta)^{\mathrm{T}}y^* - \mathbf{e}z^*] \ge 0, \forall x \ge 0 \\
(y - y^*)^{\mathrm{T}}[-(\tau - (\mu^0 - \delta)x^*)] \ge 0, \forall y \ge 0 \\
\tau - (\mu^0 - \delta)x^* \le 0 \\
\mathbf{e}^{\mathrm{T}}x^* - 1 = 0 \\
y^{*\mathrm{T}}[\tau - (\mu^0 - \delta)x^*] = 0
\end{cases}
\tag{18}
$$

By the well-known projection theorem [45], the KKT conditions can be rewritten as the following projection equations:

$$
\begin{cases}
[x - (\hat{\Sigma}x + (-\mu^0 + \delta)^{\mathrm{T}}y - \mathbf{e}z)]^+ - x = 0 \\
[y + (\tau - (\mu^0 - \delta)x)]^+ - y = 0 \\
(\mathbf{e}^{\mathrm{T}}x - 1) = 0
\end{cases}
\tag{19}
$$

That is, $w^* = (x^*, y^*, z^*)$ is a KKT point if and only if $w^* = (x^*, y^*, z^*)$ satisfies equation (19).

The following recurrent neural network is proposed in this paper for solving (7):

$$
\frac{d}{dt}\begin{pmatrix} x \\ y \\ z \end{pmatrix} = \lambda \begin{pmatrix}
-x + [x - (\hat{\Sigma}x + (-\mu^0 + \delta)^{\mathrm{T}}y - \mathbf{e}z)]^+ \\
-y + [y + (\tau - (\mu^0 - \delta)x)]^+ \\
-(\mathbf{e}^{\mathrm{T}}x - 1)
\end{pmatrix},
\tag{20}
$$

where $[y]^+ = \max\{y, 0\}$.

## IV. STABILITY ANALYSIS

In this section, we analyze the global convergence of the proposed neural network (20). First, we give a definition for use in later discussions.

**Definition 6.** $w^* = (x^*, y^*, z^*)$ *is said to be an equilibrium point of the neural network (20) if there exist $y^*$ and $z^*$ such that*

$$
\begin{cases}
0 = -x^* + [x^* - (\hat{\Sigma}x^* + (-\mu^0 + \delta)^{\mathrm{T}}y^* - \mathbf{e}z^*)]^+ \\
0 = -y^* + [y^* + (\tau - (\mu^0 - \delta)x^*)]^+ \\
0 = \mathbf{e}^{\mathrm{T}}x^* - 1
\end{cases}
$$

**Theorem 1.** *The proposed neural network of (20) with the initial point $w_0 = (x_0, y_0, z_0)$ is stable in the Lyapunov sense and globally convergent to the equilibrium point $w^* = (x^*, y^*, z^*)$.*

*Proof.* Without loss of generality, we assume $\lambda = 1$. We first show that for any initial point $w_0 = (x_0, y_0, z_0)$ with $x_0 \ge 0$ and $y_0 \ge 0$, the neural network (14) has a unique continuous solution $w(t) = (x(t), y(t), z(t))$ and $x(t) \ge 0$ and $y(t) \ge 0$.

We know that $[x - (\hat{\Sigma}x + (-\mu^0 + \delta)^{\mathrm{T}}y - \mathbf{e}z)]^+$ and $[y + (\tau - (\mu^0 - \delta)x)]^+$ are locally Lipschitz continuous. According to the local existence and uniqueness theorem of ordinary differential equations(ODEs) [46], it follows that there exists a unique continuous solution $w(t)$ of (20) for $(t_0, T)$. We can get that $w(t)$ is bounded and extend the local existence for the solution of (20) to the global existence.

According to the (20), we have the equivalent form of (20) as follows:

$$
\begin{cases}
\dfrac{dx}{dt} + x(t) = [x - (\hat{\Sigma}x + (-\mu^0 + \delta)^{\mathrm{T}}y - \mathbf{e}z)]^+ \\
\dfrac{dy}{dt} + y(t) = [y + (\tau - (\mu^0 - \delta)x)]^+ \\
\dfrac{dz}{dt} + z(t) = (\mathbf{e}^{\mathrm{T}}x - 1) + z
\end{cases}
\tag{21}
$$

Then

$$
\begin{cases}
\displaystyle\int_{t_0}^{t}\left(\frac{dx}{dt} + x(t)\right)e^s ds = \int_{t_0}^{t} e^s[x - (\hat{\Sigma}x + (-\mu^0 + \delta)^{\mathrm{T}}y \\
\qquad - \mathbf{e}z)]^+ ds \\
\displaystyle\int_{t_0}^{t}\left(\frac{dy}{dt} + y(t)\right)e^s ds = \int_{t_0}^{t} e^s[y + (\tau - (\mu^0 - \delta)x)]^+ ds \\
\displaystyle\int_{t_0}^{t}\left(\frac{dz}{dt} + z(t)\right)e^s ds = \int_{t_0}^{t} e^s((\mathbf{e}^{\mathrm{T}}x - 1) + z)ds
\end{cases}
\tag{22}
$$

On the left-hand term of (22), we get

$$
\begin{cases}
\int_{t_0}^{t}(\frac{dx}{dt}+x(t))e^s ds = \int_{t_0}^{t} e^s \frac{d[e^s x(s)]}{ds} = e^t x(t) - e^{t_0} x(t_0) \\
\int_{t_0}^{t}(\frac{dy}{dt}+y(t))e^s ds = \int_{t_0}^{t} e^s \frac{d[e^s y(s)]}{ds} = e^t y(t) - e^{t_0} y(t_0) \\
\int_{t_0}^{t}(\frac{dz}{dt}+z(t))e^s ds = \int_{t_0}^{t} e^s \frac{d[e^s z(s)]}{ds} = e^t z(t) - e^{t_0} z(t_0)
\end{cases}
\tag{23}
$$

Then

$$
\begin{cases}
x(t) = e^{-(t-t_0)} x(t_0) \\
\quad + e^{-t} \int_{t_0}^{t} e^s [x - (\hat{\Sigma} x + (-\mu^0 + \delta)^{\mathrm{T}} y - \mathbf{e} z)]^+ ds \\
y(t) = e^{-(t-t_0)} y(t_0) \\
\quad + e^{-t} \int_{t_0}^{t} e^s [y + (\tau - (\mu^0 - \delta) x)]^+ ds \\
z(t) = e^{-(t-t_0)} z(t_0) \\
\quad + e^{-t} \int_{t_0}^{t} e^s ((\mathbf{e}^{\mathrm{T}} x - 1) + z) ds
\end{cases}
\tag{24}
$$

Since $x(t_0) \geq 0$ and $[x - (\hat{\Sigma} x + (-\mu^0 + \delta)^{\mathrm{T}} y - \mathbf{e} z)]^+ \geq 0$, we have $x(t) \geq 0$. Similarly, from $y(t_0) \geq 0$ and $[y + (\tau - (\mu^0 - \delta) x)]^+$, we have $y(t) \geq 0$.

Then, the following Lyapunov function is defined:

$$
V(w) = -G(w)^{\mathrm{T}} H(w) - \frac{1}{2} \|H(w)\|_2^2 + \frac{1}{2} \|w - w^*\|_2^2, \tag{25}
$$

where

$$
G(w) = \begin{pmatrix} \hat{\Sigma} x + (-\mu^0 + \delta)^{\mathrm{T}} y - \mathbf{e} z \\ -(\tau - (\mu^0 - \delta) x) \\ -(\mathbf{e}^{\mathrm{T}} x - 1) \end{pmatrix} \tag{26}
$$

and

$$
H(w) = \begin{pmatrix} [x - (\hat{\Sigma} x + (-\mu^0 + \delta)^{\mathrm{T}} y - \mathbf{e} z)]^+ - x \\ [y + (\tau - (\mu^0 - \delta) x)]^+ - y \\ -(\mathbf{e}^{\mathrm{T}} x - 1) \end{pmatrix}. \tag{27}
$$

We can get that

$$
\frac{dV(w)}{dw} = G(w) - (\nabla G(w) - I) H(w) + (w - w^*), \tag{28}
$$

where $\nabla G(w)$ denotes the Jacobian matrix of $G$ and

$$
\nabla G(w) = \begin{bmatrix} \hat{\Sigma}^{\mathrm{T}} & -\mu^0 + \delta & -\mathbf{e}^{\mathrm{T}} \\ (\mu^0 - \delta)^{\mathrm{T}} & 0 & 0 \\ \mathbf{e} & 0 & 0 \end{bmatrix}. \tag{29}
$$

According to the Chain Rule, we can obtain that

$$
\begin{aligned}
\frac{dV(w)}{dt} &= \frac{dV(w)}{dw} \frac{dw}{dt} \\
&= [G(w) - (\nabla G(w) - I) H(w) + (w - w^*)]^{\mathrm{T}} H(w) \\
&= [G(w) + (w - w^*)]^{\mathrm{T}} H(w) - H(w)^{\mathrm{T}} \nabla G(w) H(w) \\
&\quad + \|H(w)\|_2^2
\end{aligned}
\tag{30}
$$

From [47], we have

$$
[G(w) + (w - w^*)]^{\mathrm{T}} (-G(w) - H(w)) \geq 0 \tag{31}
$$

and

$$
G(w)^{\mathrm{T}} H(w) \leq -\|H(w)\|_2^2. \tag{32}
$$

Then, we have

$$
[G(w) + (w - w^*)]^{\mathrm{T}} H(w) \leq -G(w)^{\mathrm{T}} (w - w^*) - \|H(w)\|_2^2 \tag{33}
$$

and

$$
-G(w)^{\mathrm{T}} H(w) - \|H(w)\|_2^2 \geq 0. \tag{34}
$$

Therefore, we get

$$
\begin{aligned}
\frac{dV(w)}{dt} &= \frac{dV(w)}{dw} \frac{dw}{dt} \\
&\leq -G(w)^{\mathrm{T}} (w - w^*) - \|H(w)\|_2^2 \\
&\quad - H(w)^{\mathrm{T}} \nabla G(w) H(w) + \|H(w)\|_2^2 \\
&= -G(w)^{\mathrm{T}} (w - w^*) - H(w)^{\mathrm{T}} \nabla G(w) H(w)
\end{aligned}
\tag{35}
$$

and

$$
V(w) \geq \frac{1}{2} \|w - w^*\|_2^2. \tag{36}
$$

Since $\nabla G(w) \succeq 0$, we have

$$
H(w)^{\mathrm{T}} \nabla G(w) H(w) \geq 0 \tag{37}
$$

and

$$
G(w)^{\mathrm{T}} (w - w^*) \geq 0. \tag{38}
$$

Then

$$
\frac{dV(w)}{dt} \leq -G(w)^{\mathrm{T}} (w - w^*) - H(w)^{\mathrm{T}} \nabla G(w) H(w) \leq 0. \tag{39}
$$

From (36) and (39), we can get that $V(w) \geq 0$ and

$$
\frac{dV(w)}{dt} \leq 0.
$$

It can be shown that the proposed neural network (20) is stable in the Lyapunov sense. Next we will show that the proposed neural network (20) is globally convergent to the equilibrium point $w^* = (x^*, y^*, z^*)$.

Then, for any initial point $w_0 \in \mathbb{R}^n \times \mathbb{R}^m \times \mathbb{R}^l$, the solution trajectory $\{w(t)\}$ is bounded. We next prove that $\frac{dV(w)}{dt} = 0$ if and only if $\frac{dw}{dt} = 0$. From $\frac{dw}{dt} = 0$ and Chain Rule, we can get that

$$
\frac{dV(w)}{dt} = \frac{dV(w)}{dw} \frac{dw}{dt} = 0. \tag{40}
$$

Let $\mathcal{L}$ is a largest invariant set and $\hat{w} = (\hat{x}, \hat{y}, \hat{z}) \in \mathcal{L}$. According to the invariant set theorem invariant set theorem, it follows that

$$
\frac{dV(\hat{w})}{dt} = 0 \tag{41}
$$

and

$$
G(\hat{w})^{\mathrm{T}} (\hat{w} - w^*) + H(\hat{w})^{\mathrm{T}} \nabla G(\hat{w}) H(\hat{w}) = 0 \tag{42}
$$

where

$$
\nabla G(\hat{w}) = \begin{bmatrix} \hat{\Sigma}^{\mathrm{T}} & -\mu^0 + \delta & -\mathbf{e}^{\mathrm{T}} \\ (\mu^0 - \delta)^{\mathrm{T}} & 0 & 0 \\ \mathbf{e} & 0 & 0 \end{bmatrix} \succeq 0. \tag{43}
$$

Since

$$(x^* - \hat{x})^{\mathrm{T}}(\hat{\Sigma}\hat{x} + (-\mu^0 + \delta)^{\mathrm{T}}\hat{y} - \mathbf{e}\hat{z}) \geq 0 \quad (44)$$

and

$$(\hat{x} - x^*)^{\mathrm{T}}(\hat{\Sigma}x^* + (-\mu^0 + \delta)^{\mathrm{T}}y^* - \mathbf{e}z^*)) \geq 0, \quad (45)$$

we have

$$(\hat{x} - x^*)^{\mathrm{T}}[\hat{\Sigma}x^* + (-\mu^0 + \delta)^{\mathrm{T}}y^* - \mathbf{e}z^* - (\hat{\Sigma}x^* + (-\mu^0 + \delta)^{\mathrm{T}}y^* - \mathbf{e}z^*)] \geq 0. \quad (46)$$

Due to $\nabla G(w) \succeq 0$ and $G(\hat{w})^{\mathrm{T}}(\hat{w} - w^*) \geq 0$, we have

$$(G(w) - G(w^*))^{\mathrm{T}}(\hat{w} - w^*) = 0 \quad (47)$$

and

$$H(\hat{w})^{\mathrm{T}}\nabla G(\hat{w})H(\hat{w}) = 0. \quad (48)$$

Since

$$H(\hat{w})^{\mathrm{T}}\nabla G(\hat{w})H(\hat{w})$$
$$= [(\hat{x} - (\hat{\Sigma}^{\mathrm{T}}\hat{x} + (-\mu^0 + \delta)^{\mathrm{T}}\hat{y} - \mathbf{e}\hat{z}))^+ - \hat{x}]\hat{\Sigma}^{\mathrm{T}} \quad (49)$$
$$\times [(\hat{x} - (\hat{\Sigma}^{\mathrm{T}}\hat{x} + (-\mu^0 + \delta)^{\mathrm{T}}\hat{y} - \mathbf{e}\hat{z}))^+ - \hat{x}] = 0,$$

we have

$$(\hat{x} - (\hat{\Sigma}^{\mathrm{T}}\hat{x} + (-\mu^0 + \delta)^{\mathrm{T}}\hat{y} - \mathbf{e}\hat{z}))^+ - \hat{x} = 0. \quad (50)$$

Thus, we get

$$\frac{dx}{dt} = 0. \quad (51)$$

Since $(G(w) - G(\hat{w}))^{\mathrm{T}}(w - w^*) = 0$, we get

$$(\hat{x} - x^*)^{\mathrm{T}}\nabla(\hat{\Sigma}^{\mathrm{T}}x_s)(\hat{x} - x^*) = 0, \quad (52)$$

where $x_s = \hat{x} + s(x^* - \hat{x})$. It follows that $\hat{x} = x^*$. Thus, we can get $\frac{dy}{dt} = 0$ and $\frac{dz}{dt} = 0$. Therefore, $\frac{dV(w)}{dt} = 0$ if and only if $\frac{dw}{dt} = 0$.

As a result, the proposed neural network (20) is globally convergent to the equilibrium point $w^*$.

**Theorem 2.** *The proposed neural network of (20) with the initial point $w_0 = (x_0, y_0, z_0)$ can converge to a solution within a finite time.*

*Proof.* According to Theorem 1, we know the proposed neural network (20) is globally convergent to the equilibrium point $w^* = (x^*, y^*, z^*)$.

Using the Lyapunov function $V(w)$ defined in Theorem 1, we have

$$\frac{dV(w)}{dt} \leq -\lambda G(w)^{\mathrm{T}}(w - w^*) - \lambda H(w)^{\mathrm{T}}\nabla G(w)H(w) \leq 0. \quad (53)$$

Then, for any point $w_0$ satisfying

$$\begin{cases} G(w)^{\mathrm{T}}(w - w^*) = 0 \\ H(w)^{\mathrm{T}}\nabla G(w)H(w) \geq 0 \end{cases} \quad (54)$$

$w_0$ must be an equilibrium point of (20). In terms of the given condition, the initial point $w_0$ is not an equilibrium point of (20). Then $G(w_0)^{\mathrm{T}}(w_0 - w^*) > 0$ or $H(w_0)^{\mathrm{T}}\nabla G(w_0)H(w_0) > 0$. Without loss of generality, we assume that $G(w_0)^{\mathrm{T}}(w_0 - w^*) > 0$. Since $w(t)$ is continuous, $G(w(t))^{\mathrm{T}}(w(t) - w^*)$ is also continuous.

Therefore, there exists $\tau > 0$ and $\gamma > 0$ such that $G(w(t))^{\mathrm{T}}(w(t) - w^*) \geq \gamma$ on $[t_0, \tau]$. Note that

$$V(w) \geq \frac{1}{2}\|w - w^*\|^2.$$

Then, for all $t \geq \tau$, we have

$$\|w(t) - w^*\|_2^2$$
$$\leq 2V(w(t)) \leq 2V(w(t_0)) - 2\lambda \int_{t_0}^{t}(w(s) - w^*)^{\mathrm{T}}G(w(s))ds$$
$$\leq 2V(w(t_0)) - 2\lambda \int_{t_0}^{\tau} G(w(\tau))^{\mathrm{T}}(w(\tau) - w^*)d\tau$$
$$\leq 2V(w(t_0)) - 2\lambda\gamma(\tau - t_0). \quad (55)$$

It can be seen that when $\lambda = 2V(w(t_0))/2\lambda\gamma(\tau - t_0)$

$$\|w(t) - w^*\|_2^2 \leq 2V(w(t_0)) - 2V(w(t_0)) = 0, \quad \forall t \geq \tau. \quad (56)$$

Thus, $w(t)$ reaches $w^*$ for all $t \geq \tau$.

## V. SIMULATION RESULTS

In this section, we demonstrate the effectiveness of the proposed approach by using some stock market data to solve the robust portfolio problem.

*Example 1:* Dow Jones Industrial Average (DJI), is one of the oldest and well-known stock market indices in the United States. It consists of 30 major U.S. companies that are influential and representative of their respective industries. The following example considers five DJI stocks, namely American Express Company (AXP), General Electric (GE), McDonald's Co. (McD), Merck & Co. Inc. (MRK), and AT&T Inc. (AT&T). The analysis is based on data from the year 2006. The mean return and variance matrix data for the five stocks are derived from [48].

We consider the following robust portfolio selection optimization problem:

$$\min \frac{1}{2}x^{\mathrm{T}}\hat{\Sigma}x$$
$$\text{s.t. } \tilde{\mu}_i x \geq \tau, \quad i = 1, \ldots, 5$$
$$\mathbf{e}^{\mathrm{T}}x = 1$$
$$x \geq 0$$
$$\tilde{\mu}_i \in \mathcal{U}_i.$$

Since the five stocks have different average returns and risks, we design five box uncertainty sets corresponding to different stocks:

$$\mathcal{U}_1 = \{\tilde{\mu}_1 : |\tilde{\mu}_1 - \mu_1^0| \leq \delta_1\},$$
$$\mathcal{U}_2 = \{\tilde{\mu}_2 : |\tilde{\mu}_2 - \mu_2^0| \leq \delta_2\},$$
$$\mathcal{U}_3 = \{\tilde{\mu}_3 : |\tilde{\mu}_3 - \mu_3^0| \leq \delta_3\},$$
$$\mathcal{U}_4 = \{\tilde{\mu}_4 : |\tilde{\mu}_4 - \mu_4^0| \leq \delta_4\},$$
$$\mathcal{U}_5 = \{\tilde{\mu}_5 : |\tilde{\mu}_5 - \mu_5^0| \leq \delta_5\}.$$

where the perturbation of the uncertain parameter $\delta \in \mathbb{R}^5$ is

$$\delta = \begin{bmatrix} 0.0002 & 0.0001 & 0.0004 & 0.0005 & 0.0008 \end{bmatrix}^{\mathrm{T}}.$$

By introducing the box uncertainty sets, we can eliminate the above uncertain parameters, and according to the above derivation, the corresponding robust counterpart is obtained:

$$\min \frac{1}{2} x^{\mathrm{T}} \hat{\Sigma} x$$
$$\text{s.t. } \tau - (\mu_i^0 - \delta_i) x \leq 0, \quad i = 1, \ldots, 5$$
$$\mathbf{e}^{\mathrm{T}} x - 1 = 0$$
$$x \geq 0.$$

Then we propose the following neural network:

$$\frac{d}{dt} \begin{pmatrix} x \\ y \\ z \end{pmatrix} = \lambda \begin{pmatrix} -x + [x - (\hat{\Sigma}x + (-\mu^0 + \delta)^{\mathrm{T}} y - \mathbf{e}z)]^+ \\ -y + [y + (\tau - (\mu^0 - \delta)x)]^+ \\ -(\mathbf{e}^{\mathrm{T}} x - 1) \end{pmatrix},$$

where the mean return $\mu_0 \in \mathbb{R}^5$ is

$$\mu^0 = \begin{bmatrix} 0.0007 & 0.0004 & 0.0013 & 0.0014 & 0.0017 \end{bmatrix}^{\mathrm{T}}.$$

and the variance matrix $\hat{\Sigma}$ is

$$\hat{\Sigma} = 10^{-3} \times \begin{bmatrix} 0.0970 & 0.0361 & 0.0376 & 0.0283 & 0.0341 \\ 0.0361 & 0.0619 & 0.0257 & 0.0230 & 0.0229 \\ 0.0376 & 0.0257 & 0.1264 & 0.0321 & 0.0254 \\ 0.0283 & 0.0230 & 0.0321 & 0.1413 & 0.0436 \\ 0.0341 & 0.0229 & 0.0254 & 0.0436 & 0.1138 \end{bmatrix}.$$

At the same time, we set the value of the expected total return $\tau = 0.0009$ and the neural network parameter $\lambda = 1000$.

Fig. 1 shows that the outputs of the neural network are convergent to a unique optimal solution $x^* = (0, 0, 0.3648, 0.2520, 0.3832)^{\mathrm{T}}$ from any initial point $x_0$. It implies the selection of stocks 3 (McD), 4 (MRK), and 5 (AT&T) for the optimal investment of a robust portfolio selection problem.

*Example 2:* In this example, we use the same model as in Example 1 along with the mean return and variance matrices. Then, we set different values for other parameters to further validate the effectiveness of the proposed method.

The perturbation of the uncertain parameter $\delta$ is

$$\delta = \begin{bmatrix} 0.0001 & 0.0003 & 0.0003 & 0.0003 & 0.0006 \end{bmatrix}^{\mathrm{T}}.$$

We design five box uncertainty sets corresponding to different stocks:

$$\mathcal{U}_1 = \{\tilde{\mu}_1 : |\tilde{\mu}_1 - 0.0007| \leq 0.0001\},$$
$$\mathcal{U}_2 = \{\tilde{\mu}_2 : |\tilde{\mu}_2 - 0.0004| \leq 0.0003\},$$
$$\mathcal{U}_3 = \{\tilde{\mu}_3 : |\tilde{\mu}_3 - 0.0013| \leq 0.0003\},$$
$$\mathcal{U}_4 = \{\tilde{\mu}_4 : |\tilde{\mu}_4 - 0.0014| \leq 0.0003\},$$
$$\mathcal{U}_5 = \{\tilde{\mu}_5 : |\tilde{\mu}_5 - 0.0017| \leq 0.0006\}.$$

At the same time, we set the value of the expected total return $\tau = 0.0009$ and the parameter of the neural network $\lambda = 1000$.

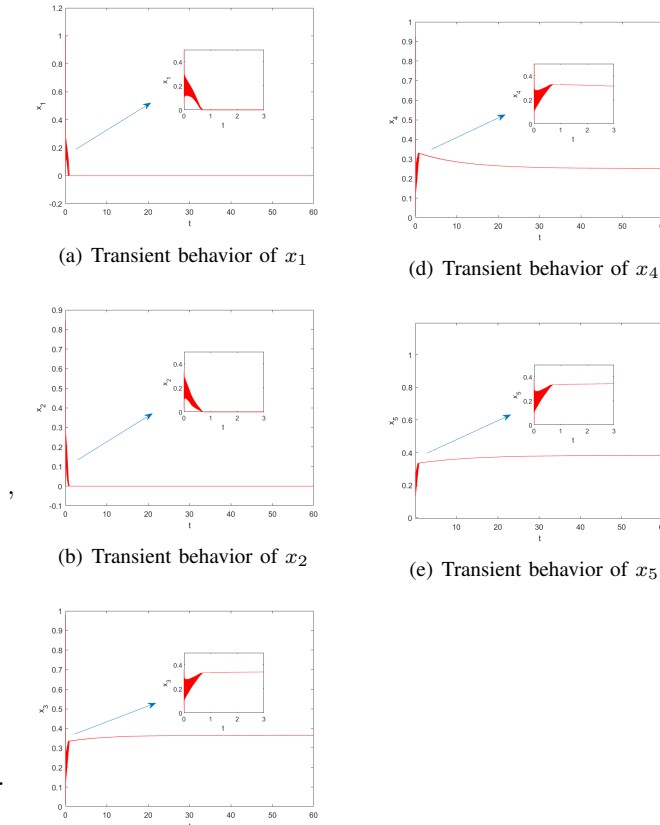

(a) Transient behavior of $x_1$

(b) Transient behavior of $x_2$

(c) Transient behavior of $x_3$

(d) Transient behavior of $x_4$

(e) Transient behavior of $x_5$

Fig. 1: Transient behaviors of the state variables of neural network (20) in Example 1.

Fig. 2 shows that the outputs of the neural network are convergent to a unique optimal solution $x^* = (0, 0, 0, 0.4192, 0.5808)^{\mathrm{T}}$ from any initial point $x_0$. It implies the selection of 4 (MRK), and 5 (AT&T) for the optimal investment of a robust portfolio selection problem.

*Example 3:* As in Example 2, we are also changing the upper and lower bounds of the box uncertainty set here.

The perturbation of the uncertain parameter $\delta$ is

$$\delta = \begin{bmatrix} 0.0003 & 0.0001 & 0.0004 & 0.0005 & 0.0006 \end{bmatrix}^{\mathrm{T}}.$$

We design five box uncertainty sets corresponding to different stocks:

$$\mathcal{U}_1 = \{\tilde{\mu}_1 : |\tilde{\mu}_1 - 0.0007| \leq 0.0003\},$$
$$\mathcal{U}_2 = \{\tilde{\mu}_2 : |\tilde{\mu}_2 - 0.0004| \leq 0.0001\},$$
$$\mathcal{U}_3 = \{\tilde{\mu}_3 : |\tilde{\mu}_3 - 0.0013| \leq 0.0004\},$$
$$\mathcal{U}_4 = \{\tilde{\mu}_4 : |\tilde{\mu}_4 - 0.0014| \leq 0.0005\},$$
$$\mathcal{U}_5 = \{\tilde{\mu}_5 : |\tilde{\mu}_5 - 0.0017| \leq 0.0006\}.$$

At the same time, we set the value of the expected total return $\tau = 0.0009$ and the parameter of the neural network $\lambda = 1000$.

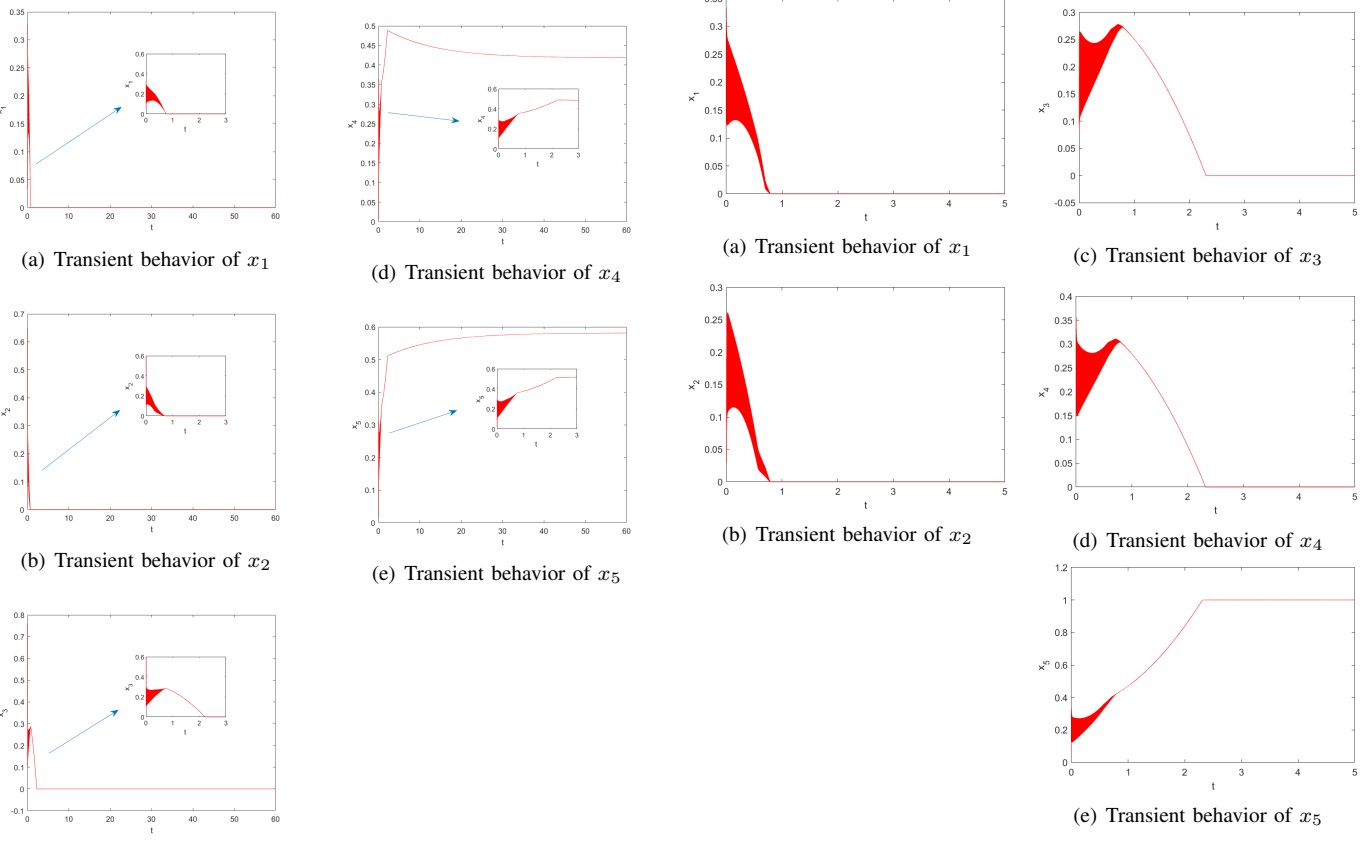

(a) Transient behavior of $x_1$

(d) Transient behavior of $x_4$

(b) Transient behavior of $x_2$

(e) Transient behavior of $x_5$

(c) Transient behavior of $x_3$

Fig. 2: Transient behaviors of the state variables of neural network (20) in Example 2.

(a) Transient behavior of $x_1$

(c) Transient behavior of $x_3$

(b) Transient behavior of $x_2$

(d) Transient behavior of $x_4$

(e) Transient behavior of $x_5$

Fig. 3: Transient behaviors of the state variables of neural network (20) in Example 3.

Fig. 3 shows that the outputs of the neural network are convergent to a unique optimal solution $x^* = (0, 0, 0, 0, 1)^{\mathrm{T}}$ from any initial point $x_0$. It implies the selection of 5 (AT&T) for optimal investment of robust portfolio selection problem.

For these three examples, three different asset weight allocations were obtained based on three different box uncertainty sets, resulting in different risk values. See Table I for details.

TABLE I: Risk Values for Different Asset Allocations

| Example Number | Asset Weight Configuration | Risk Value ($\frac{1}{2}x^{\mathrm{T}}\hat{\Sigma}x$) |
|---|---|---|
| 1 | [0, 0, 0.3648, 0.2520, 0.3832] | 0.00000697 |
| 2 | [0, 0, 0, 0.4192, 0.5808] | 0.00004582 |
| 3 | [0, 0, 0, 0, 1] | 0.00001288 |

The results demonstrate that a diversified asset allocation (Example 1) outperforms concentrated investment strategies (Examples 2 and 3) with respect to risk control, exhibiting a markedly lower value-at-risk (VaR). This phenomenon can be attributed to the fact that diversification serves to mitigate the sensitivity of the portfolio to the volatility of a single asset. Conversely, a concentrated investment strategy may offer the potential for higher expected returns in certain market conditions. However, this approach involves an increased

risk exposure in uncertain or extreme market environments. The experimental results demonstrate that the neurodynamic approach exhibits rapid convergence and efficiency in the resolution of optimization problems.

*Example 4:* This example is based on the HDAX ((Deutsche Borse) dataset, which is obtained from the work of [27]. The data set is constructed based on the 49 adjusted weekly closing prices of stocks from January 3, 2000, to December 29, 2017.

We randomly selected 20 stocks to demonstrate the effectiveness of the neural network. In addition, we derived the weekly returns and variance of the 20 stocks.

The mean return $\mu_0 \in \mathbb{R}^{20}$ is a vector and

$$\mu^0 = [0.1546 \quad -0.0037 \quad -0.0294 \quad 0.0656 \quad 0.0855$$
$$0.0605 \quad -0.2256 \quad 0.1973 \quad 0.0293 \quad -0.0250$$
$$0.0023 \quad 0.0667 \quad -0.0110 \quad 0.0026 \quad 0.0661$$
$$0.0428 \quad 0.0222 \quad 0.0254 \quad 0.0406 \quad -0.0256]^{\mathrm{T}}$$

Similarly, we can obtain the variance matrix $\hat{\Sigma} \in \mathbb{R}^{20 \times 20}$.

We consider the following robust portfolio selection opti-

mization problem:

$$\min \frac{1}{2}x^{\mathrm{T}}\hat{\Sigma}x$$
$$\text{s.t. } \tilde{\mu}_i x \geq \tau, \quad i = 1, \ldots, 20$$
$$\mathbf{e}^{\mathrm{T}}x = 1$$
$$x \geq 0$$
$$\tilde{\mu}_i \in \mathcal{U}_i.$$

Since the 20 stocks have different average returns and risks, we present 20 box uncertainty sets with respect to different stocks:

$$\mathcal{U}_1 = \{\tilde{\mu}_1 : |\tilde{\mu}_1 - \mu_1^0| \leq \delta_1\},$$
$$\mathcal{U}_2 = \{\tilde{\mu}_2 : |\tilde{\mu}_2 - \mu_2^0| \leq \delta_2\},$$
$$\ldots\ldots$$
$$\mathcal{U}_{20} = \{\tilde{\mu}_{20} : |\tilde{\mu}_{20} - \mu_{20}^0| \leq \delta_{20}\},$$

where the perturbation of the uncertain parameter $\delta \in \mathbb{R}^{20}$ is a vector and

$$\delta = [0.1346 \quad 0.0017 \quad 0.0094 \quad 0.0456 \quad 0.0655$$
$$0.0405 \quad 0.2056 \quad 0.1773 \quad 0.0093 \quad 0.0050$$
$$0.0003 \quad 0.0467 \quad 0.0010 \quad 0.0006 \quad 0.0461$$
$$0.0228 \quad 0.0022 \quad 0.0054 \quad 0.0206 \quad 0.0056]^{\mathrm{T}}$$

By introducing the box uncertainty sets, we can eliminate the above uncertain parameters, and according to the above derivation, the corresponding robust counterpart is obtained:

$$\min \frac{1}{2}x^{\mathrm{T}}\hat{\Sigma}x$$
$$\text{s.t. } \tau - (\mu_i^0 - \delta_i)x \leq 0, \quad i = 1, \ldots, 20$$
$$\mathbf{e}^{\mathrm{T}}x - 1 = 0$$
$$x \geq 0.$$

Then we propose the following neural network:

$$\frac{d}{dt}\begin{pmatrix} x \\ y \\ z \end{pmatrix} = \lambda \begin{pmatrix} -x + [x - (\hat{\Sigma}x + (-\mu^0 + \delta)^{\mathrm{T}}y - \mathbf{e}z)]^+ \\ -y + [y + (\tau - (\mu^0 - \delta)x)]^+ \\ -(\mathbf{e}^{\mathrm{T}}x - 1) \end{pmatrix}$$

where $[y]^+ = \max\{y, 0\}$. At the same time, we set the value of the expected total return $\tau = 0.002$ and the parameter of the neural network $\lambda = 0.1$.

Fig. 4 shows that the outputs of the neural network are convergent to a unique optimal solution $x^* = [3.166 \times 10^{-46}, 3.52 \times 10^{-46}, 0.005946, 3.55 \times 10^{-46}, 2.458 \times 10^{-46}, 3.791 \times 10^{-47}, 1.082 \times 10^{-46}, 2.125 \times 10^{-46}, 3.721 \times 10^{-46}, 3.75 \times 10^{-46}, 0.6884, 3.772 \times 10^{-46}, 0.1305, 0.1469, 6.643 \times 10^{-45}, 1.099 \times 10^{-29}, 9.356 \times 10^{-45}, 3.559 \times 10^{-46}, 6.407 \times 10^{-42}, 0.02816]$, from any initial point $x_0$. At the same time, the objective function can be calculated as $\frac{1}{2}x^{\mathrm{T}}\hat{\Sigma}x = 0.0229$. The solution shows the selection of the 3rd, 11th, 13th, 14th, and 20th stocks with weights [0.005946, 0.6884, 0.1305, 0.1469, 0.02816].

The simulation results show that the proposed neural network can adapt to market volatility and find effective portfolios. It also provides reliable investment decisions on global stock markets. The results demonstrate the effectiveness of the proposed neural network.

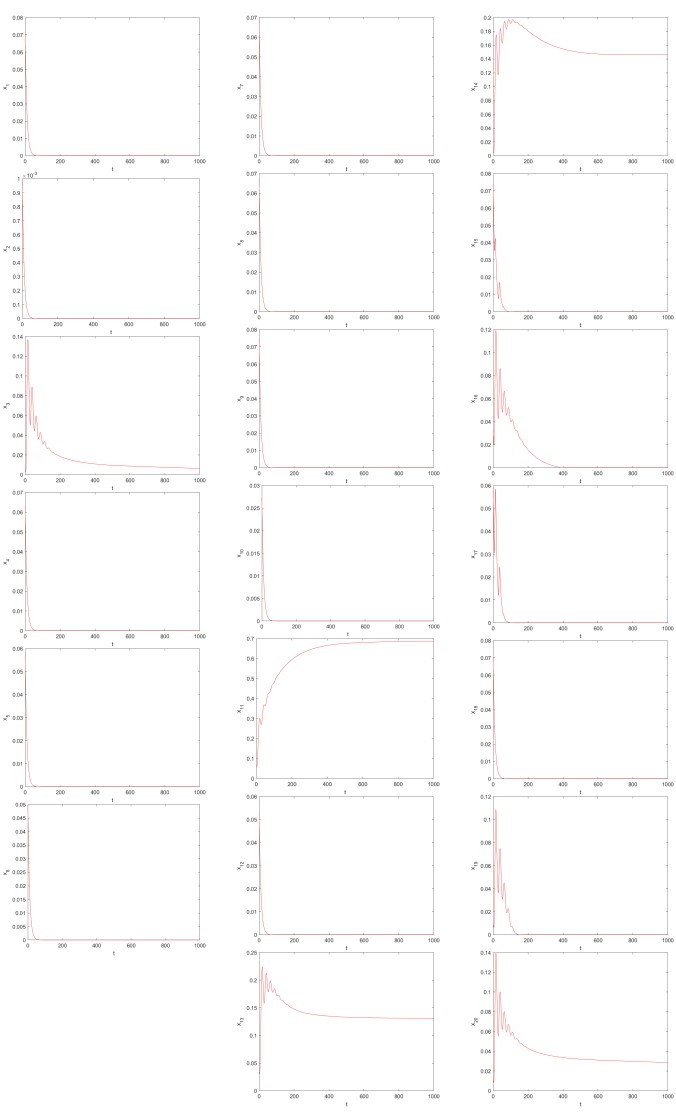

Fig. 4: Transient behaviors of the state variables of neural network (20) in Example 4.

## VI. CONCLUSION

In this paper, we propose a neural network for solving robust portfolio selection optimization problems under the box uncertainty set. By eliminating the uncertainties, we obtain the robust counterpart model, and the set of equilibrium points of the proposed neural network is equivalent to the set of KKT optimal points of the robust counterpart model. Each equilibrium point of the proposed neural network is stable in the Lyapunov sense, and the state of the proposed neural network converges to an equilibrium point from any initial point. Simulation results demonstrate the global convergence and effectiveness of the proposed neural network. Further research is to design new neural networks to solve robust portfolio selection optimization problems based on different uncertainty sets and apply them to solve more practical problems.

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
