# OpenReview forum: "A One-layer Neural Network for Robust Mean-Variance Portfolio Selection Problem"
_IEEE.org/ICIST/2024/Conference — IEEE ICIST 2024 Conference Submission_

### Official Review · Reviewer_fuRx · 2024-08-22
**This article is quite fascinating and of high quality.**

**Rating:** 7
**Confidence:** 3

**Review:**

This paper, " A One-layer Neural Network for Robust Mean-Variance Portfolio Selection Problem," proposes a neurodynamic approach to robust portfolio selection. Firstly, the uncertainty of the robust portfolio model under the uncertainty set of the box is eliminated, and the corresponding robust pairing model is derived. Finally, a single layer neural network model is constructed based on Karush-Kuhn-Tucker (KKT) condition. The article has clear logic and organization, but there are still some problems. My specific feedback is as follows :1) In the abstract part, the author should briefly discuss the problem in the first two sentences. 2) In the introduction, the author has insufficient background content for the study of UAV path planning. 2) In the research, what are the advantages of KKT conditional neural network model compared with general neural network model?

---

### Official Review · Reviewer_s7z6 · 2024-08-24
**This paper introduced a neurodynamic approach to robust portfolio selection. This approach is capable of efficiently handling high-dimensional data through massively parallel processing, providing a resilient solution to the complexities of modern financial markets. The topic of this paper is interesting. Below is a list of comments that should be taken into account further when revising the paper.**

**Rating:** 7
**Confidence:** 3

**Review:**

1. The paper should provide a detailed description of the innovative points to enable readers to quickly understand the paper. At the same time, it makes the structure of the article more complete.
2. In the preliminaries section, the author should number the formulas for definitions and lemmas, which makes the framework of the article clearer.
3. In the simulation results section, the author should summarize the four examples to obtain the global convergence and effectiveness of the neural network proposed in this paper.

---

### Official Review · Reviewer_Fd4C · 2024-08-25
**Accept**

**Rating:** 7
**Confidence:** 3

**Review:**

Comment: This paper introduces a neurodynamic approach to robust portfolio selection and simulation experiments are conducted using two global stock market datasets. The theory is correct and can be accepted after responding the following comments.
(1) In the introduction, it is not enough to state the current work. It should be expended and reconstructed.
(2) What is the contribution of the paper? It should be highlighted both in the introduction and in the content.
(3) In the simulation section, it is recommended to compare it with some previous methods.

---

### Decision · Program_Chairs · 2024-09-06

Accept (Oral)